# Interscapular and Perivascular Brown Adipose Tissue Respond Differently to a Short-Term High-Fat Diet

**DOI:** 10.3390/nu11051065

**Published:** 2019-05-13

**Authors:** Peter Aldiss, Jo E. Lewis, David J. Boocock, Amanda K. Miles, Ian Bloor, Francis J. P. Ebling, Helen Budge, Michael E. Symonds

**Affiliations:** 1The Early Life Research Unit, Division of Child Health, Obstetrics and Gynaecology, School of Medicine, University of Nottingham, Nottingham NG7 2UH, UK; peter.aldiss@nottingham.ac.uk (P.A.); ian.bloor@nottingham.ac.uk (I.B.); helen.budge@nottingham.ac.uk (H.B.); 2School of Life Sciences, Queen’s Medical Centre, University of Nottingham, Nottingham NG7 2UH, UK; jl2033@medschl.cam.ac.uk (J.E.L.); fran.ebling@nottingham.ac.uk (F.J.P.E.); 3John van Geest Cancer Research Centre, Nottingham Trent University, Nottingham NG11 8NS, UK; david.boocock@ntu.ac.uk (D.J.B.); amanda.miles@ntu.ac.uk (A.K.M.); 4Nottingham Digestive Disease Centre and Biomedical Research Unit, School of Medicine, University of Nottingham, Nottingham NG7 2UH, UK

**Keywords:** brown fat, white fat, proteome, nutrient excess

## Abstract

Brown adipose tissue (BAT) function may depend on its anatomical location and developmental origin. Interscapular BAT (iBAT) regulates acute macronutrient metabolism, whilst perivascular BAT (PVAT) regulates vascular function. Although phenotypically similar, whether these depots respond differently to acute nutrient excess is unclear. Given their distinct anatomical locations and developmental origins and we hypothesised that iBAT and PVAT would respond differently to brief period of nutrient excess. Sprague-Dawley rats aged 12 weeks (n=12) were fed either a standard (10% fat, n=6) or high fat diet (HFD: 45% fat, n=6) for 72h and housed at thermoneutrality. Following an assessment of whole body physiology, fat was collected from both depots for analysis of gene expression and the proteome. HFD consumption for 72h induced rapid weight gain (c. 2.6%) and reduced serum non-esterified fatty acids (NEFA) with no change in either total adipose or depot mass. In iBAT, an upregulation of genes involved in insulin signalling and lipid metabolism was accompanied by enrichment of lipid-related processes and functions, plus glucagon and peroxisome proliferator-activated receptor (PPAR) signalling pathways. In PVAT, HFD induced a pronounced down-regulation of multiple metabolic pathways which was accompanied with increased abundance of proteins involved in apoptosis (e.g., Hdgf and Ywaq) and toll-like receptor signalling (Ube2n). There was also an enrichment of DNA-related processes and functions (e.g., nucleosome assembly and histone exchange) and RNA degradation and cell adhesion pathways. In conclusion, we show that iBAT and PVAT elicit divergent responses to short-term nutrient excess highlighting early adaptations in these depots before changes in fat mass.

## 1. Introduction

Adipose tissue function differs with its anatomical location and developmental origin [1]. For instance, whilst interscapular brown adipose tissue (iBAT) shares its lineage with skeletal muscle (e.g., Myf5+), perivascular brown adipose tissue (PVAT) is thought to derive from vascular smooth muscle cells (e.g., SM22α+) [1,2]. iBAT can play a role in whole body glucose, and lipid homeostasis as well as thermoregulation through the activation of uncoupling protein 1 (UCP1) which dissipates chemical energy as heat bypassing the conversion of ADP to ATP [3,4,5]. Despite PVAT being phenotypically similar to iBAT, i.e., abundant in UCP1 and other thermogenic genes, its primary physiological role is the regulation of vascular function rather than systemic metabolism per se [2]. Dysfunctional BAT may contribute to obesity and associated metabolic disease, whereas compromised PVAT may enhance the atherogenic processes due to its close proximity to and crosstalk with the endothelium [6,7].

The effect of diet-induced obesity on BAT is well established [8], but less is known on its response to brief periods of high-fat feeding. Central inflammation occurs after only 3-4 days of commencing a high-fat diet (HFD) that is accompanied with central and peripheral insulin resistance, adipose tissue inflammation and hepatic steatosis [9,10,11,12,13,14,15,16]. In humans, insulin resistance can be induced after 24h of a saturated-fatty acid (SFA) rich diet, with longer periods of overfeeding causing similar results to those seen in animal models [17]. Although both iBAT and PVAT contain abundant UCP1 and express glycolytic/lipolytic genes [18], their response to brief nutrient excess is unclear. Therefore, we determined whether iBAT and PVAT differ in their response to a short-term (i.e., 72h) caloric surplus. Given the evidence that ambient housing temperature is a critical factor in determining adipose tissue function [19,20], we determined the response to a HFD at thermoneutrality (Tn, 28-30°C) so as to mimic human physiology by studying BAT under basal conditions (i.e., when UCP1 is not active).

## 2. Materials and Methods

All studies were approved by the University of Nottingham Animal Welfare and Ethical Review Board, and were carried out in accordance with the UK Animals (Scientific Procedures) Act of 1986. Twelve male Sprague-Dawley rats aged 8 weeks were purchased from Charles River (Kent, UK). Animals were randomised (http://www.graphpad.com/quickcalcs/randomize1.cfm) to either the control or intervention group. The study was carried out at thermoneutrality (c. 28⁰C) to negate any confounding effects of active BAT on the response to nutrient excess, and animals were acclimated to this environment for 4 weeks. Following the 4 week acclimation, all animals were weighed and received either the control diet (824050 SDS, Kent, UK) or a 45% high-fat (HFD, n=6) diet (824018 SDS, Kent, UK) for 72h. During this time, animals had ad libitum access to food and water and all procedures were carried out under a 12:12-hour reverse light-dark cycle (i.e., the during the active phase) so as to minimise animal stress and maximise data quality and translatability [21].

### 2.1. Metabolic Cages

All animals were placed in an open-circuit calorimeter known as the ‘comprehensive laboratory animal monitoring system’ (CLAMS: Columbus Instruments, Linton Instrumentation, Diss, UK) for the last 48h of the study. Oxygen consumption (VO_2_) and carbon dioxide production (VCO_2_) were measured [22] and were then used to calculate energy expenditure (EE) and respiratory exchange ratio (RER) [23,24], as previously described. Measurements were taken at 9 minute intervals for the last 24h. At the end of the 24h period, all animals were weighed and fasted overnight prior to euthanasia by rising CO_2_ gradient. Relevant tissues were then rapidly dissected, weighed, snap-frozen in liquid nitrogen and stored at -80°C for subsequent analysis. PVAT was dissected from the aortic arch down the thoracic aorta [25].

### 2.2. Gene Expression Analysis

Total RNA was extracted from iBAT and PVAT with the RNeasy Plus Micro extraction kit (Qiagen, West Sussex, UK) using an adapted version of the single step acidified phenol-chloroform method. RNA purity was subsequently quantified with the Nanodrop ND-100 (Nanodrop Technologies, Wilmington, USA) and all samples were normalised to 1 ng μL^−1^. Reverse transcription was carried out using the High Capacity RNA-to-cDNA kit (Life Technologies, Paisley, UK) and cDNA was then amplified on a Touchgene Gradient thermocycler (Techne Inc, Bibby Scientific Limited, Staffordshire, UK). Genes regulating thermogenesis, insulin signalling and energy metabolism were analysed by quantitative PCR on the Step One Plus q-PCR system and v.2.2 software (Applied Biosciences) using either iTaq Universal SybrGreen mastermix (BioRad, Watford, UK) or Taqman universal mastermix (ThermoFisher, Loughborough, UK) with rat-specific oligonucleotide primers (Sigma, Gillingham, UK) or FAM-MGB Taqman probes (see Appendix A for primer list). Gene expression was determined using the GeNorm algorithm against two selected reference genes: *YWHAZ* and *TBP* (stability value M = 0.18 in BAT and 0.25 in PVAT).

### 2.3. Targeted Insulin Resistance PCR Arrays

We utilised the Insulin Resistance (SAB target list) PCR Array (BioRad) to screen for 86 genes involved in the onset of adipose tissue insulin resistance (n=3 per group). All procedures were carried out according to manufacturers’ instructions. Validation of representative data is shown in Appendix A.

### 2.4. Serum Analysis

Serum was thawed gently on ice with concentrations of glucose (GAGO-20, Sigma Aldrich, Gillingham, UK), triglycerides (LabAssay ™ Trigylceride, Wako, Neuss, Germany), non-esterified fatty acids (NEFA)-HR(2), (Wako) and insulin (80-INSRT-E01, Alpco, Salem, NH, USA) measured following manufacturer’s instructions.

### 2.5. Protein Extraction, Clean-Up and Trypsinization

Proteins were extracted by homogenisation of c. 50-100 mg of frozen tissue in 500 μL CellLytic MT cell lysis buffer (Sigma, C3228) and 5 μL of Halt Protease Inhibitor Cocktail (Thermo, 78430) with subsequent centrifugation at 20,000× *g* for 10 min. The concentration of each supernatant was determined using the Pierce BCA Protein Assay Kit (Thermo, 23225) prior to storage at -80°C. Lipid and other contaminants were removed from 100 μL of each protein lysate using the ReadyPrep 2D cleanup Kit (Biorad, 1632130) with the final protein pellet reconstituted in 100 μL of 50 mM TEAB buffer (6 M Urea, pH 8.0). Following quantification of the post-clean up concentration each sample was normalised (50 ug) and 5 μL of 200 mM DTT/50 mM TEAB (pH 8.0) was added to each for the reduction of proteins over a 1 h period. Following this, 20 μL of 200 mM Iodoacetamide/50 mM TEAB (pH 8.0) was added for alkylation (1 h) and finally, 20 μL of 200 mM DTT/50 mM TEAB (pH 8.0) to consume unreacted Iodoacetamide (1 h) with the latter two incubations carried out in the dark. 775 μL of 50 mM TEAB was then added to reduce the urea concentration to c. 0.6 M and Sequencing Grade Modified Trypsin (Promega, V5113) solution was added in a final concentration of 1:20 (w:w trypsin/protein). All samples were gently vortexed and incubated overnight for 18 h at 37°C, following which 2.5 μL of formic acid was added to reduce the pH and halt trypsin activity. All samples were then dried down at 60°C for 4 h and stored at 80°C before resuspending in liquid chromatography mass spectrometry (LCMS) grade 5% acetonitrile in 0.1% formic acid for subsequent analysis.

### 2.6. Mass Spectrometry

Samples (4 μL) were injected by Eksigent 425 LC system onto a trap column (Mobile Phase A; 0.1% formic acid, B; Acetonitrile with 0.1% formic acid; YMC Triart C_18_ guard column 0.3 × 5 mm, 300 μm ID) at 10 μL/min mobile phase A for 2 min before gradient elution onto the analytical column (YMC Triart C_18_ 150 × 0.3mm ID, 3 μm) in line to a Sciex TripleTOF 6600 Duospray Source using a 50 μm electrode, positive mode +5500V. Samples were analysed in both IDA (Information Dependent Acquisition, for the generation of a spectral library) and SWATH (Data Independent Acquisition, to generate quantitative data) modes. The following linear gradients were used: for IDA, mobile phase B increasing from 2% to 30% over 68 min; 40% B at 72 min followed by column wash at 80% B and re-equilibration (87 min total run time). For SWATH, 3-30% B over 38 min; 40% B at 43 min followed by wash and re-equilibration as before (57 min total run time). IDA acquisition mode was used with a top 30 ion fragmentation (TOFMS *m/z* 400-1250; product ion 100-1500) followed by 15 sec exclusion using rolling collision energy, 50 ms accumulation time; 1.8 s cycle. SWATH acquisition was using 100 variable windows (optimised on sample type) 25 ms accumulation time, 2.6 s cycle (*m/z* 400-1250). IDA data was searched together using ProteinPilot 5.0.2, iodoacetamide alkylation, thorough search with emphasis on biological modifications (Swissprot rat database June 2018). SWATH data was analysed using Sciex OneOmics software [26] extracted against the locally generated library with the parameters 12 peptides per protein, 6 transitions per peptide, XIC width 30 ppm, 5 min retention time window.

### 2.7. Statistical Analysis

Statistical analysis was performed in GraphPad Prism version 8.0 (GraphPad Software, San Diego, CA). Data are expressed as Mean ± SEM and details of specific statistical tests are given in figure legends.

Functional analysis of the proteome was performed using the Advaita Bioinformatic iPathwayGuide software (www.advaitabio.com/ipathwayguide.html) with a fold change ± 0.5 and confidence score cut-off of 0.75. Significantly impacted biological processes, molecular interactions and pathways were analysed in the context of pathways obtained from the Kyoto Encyclopedia of Genes and Genomes (KEGG) database (Release 84.0+/10-26, Oct 17) [27] and the Gene Ontology Consortium database (2017-Nov) [28]. The Elim pruning method, which removes genes mapped to a significant gene ontology (GO) term from more general (higher level) GO terms, was used to overcome the limitation of errors introduced by considering genes multiple times [29].

## 3. Results

### 3.1. Short-Term HFD Downregulates Genes Involved in Thermogenesis in PVAT Only

As expected, the short period of consuming a HFD did not cause any difference in final body weight between groups (Figure 1A), or in total fat mass (Control: 22.87±3.31; HFD: 23.51±3.68 g) or the weight of individual fat depots (Appendix A). Increased 24h energy intake (Figure 1B) led to a small but significant increase final body weight (Figure 1C) equal to c. 2.6% body weight. Whilst there was no change in ambulatory activity or energy expenditure (Figure 1D,F, Appendix A) the reduction in RER (Figure 1E) reflects a shift towards fat as the major fuel substrate in the HFD group. Interestingly, despite this rapid weight gain, there were no differences in serum insulin, glucose or triglycerides, although NEFA was reduced (Figure 1G–J). Thermogenic genes in BAT were unaffected, whereas PVAT was more susceptible to the HFD (Figure 2A,C). Despite similar UCP1 messenger RNA (mRNA), there was a reduction in gene expression for β3AR, DIO2 and PRDM16 in PVAT. There were no differences with the HFD in PGC1α, CIDEA, FGF21, CITED1, SLC36a2 or P2RX5 in iBAT, whilst only SLC36a2 was reduced in PVAT. Markers of beige adipocytes, TBX1 and TMEM26, were expressed in both iBAT and PVAT and the HFD only reduced TMEM26 in iBAT. The white adipocyte specific cell-surface marker, ASC1 [30], was expressed in both iBAT and PVAT and reduced with HFD in the latter. 

### 3.2. Short-Term HFD Alters Insulin Signalling Pathways in a Depot-Specific Manner

Due to the marked reduction in thermogenic genes in PVAT in response to the HFD, a number of genes that regulate insulin signalling and energy metabolism were measured to determine if these were affected by short-term nutrient excess. Targeted array profiling demonstrated that genes involved in insulin signalling (i.e., Igf1, Insr, and Mapk3) and lipid metabolism (i.e., Hsl, Lpl, Acsl1 and Srebf2) were upregulated in iBAT (Figure 2B,D). iBAT also exhibited increased expression of genes associated with ‘whitening’ (i.e., Lep, Retn and Adipoq) and inflammation (i.e., Tlr4, Emr1 and Tnfrsf1b). Only four genes were down-regulated in iBAT including Pdk2 and Il6. PVAT exhibited a marked upregulation of Ccr4 and Cxcr3 which govern T-cell differentiation and infiltration. The HFD increased in markers of inflammasome activation, Pycard, Il1β and Casp1 in PVAT, concomitant with a down-regulation of genes governing lipid (i.e., Hsl, Pde3B, Acacb and Ppara) and glucose metabolism (i.e., Cs, Gys1, Irs2 and mTOR).

### 3.3. Short-Term High Fat Feeding Induces Divergent Alterations in the BAT and PVAT Proteome

As the HFD induced clear differences in depot response, we analysed the proteome to identify novel proteins and pathways regulated by short-term high-fat feeding. A total of 107 proteins were differentially regulated in iBAT with those involved in the 20S core proteasome complex (Psma3l, ↑), endocytosis (Vps4a, ↓), calcium signalling (Camk2d, ↓) and glycolysis (Pkm, ↑) amongst the most significant (Table 1). In PVAT, 183 proteins were differentially regulated including those involved in glycolysis (Pfkp, ↓), apoptosis (Hdgf,↑; Ywhaq, ↑ and Ghitm, ↓), TLR signalling (Ube2n, ↑) and peroxisomal lipid metabolism (Acox1, ↓) (Table 2).

Gene ontology (GO) analysis demonstrated the proteins in iBAT (Table 3) were significantly enriched for lipid-related processes and functions including *positive regulation of lipid catabolic process, high-density lipoprotein particle assembly, phosphatidylcholine-sterol O-acyltransferase activator activity* and *very-low-density lipoprotein particle*. In PVAT (Table 4), however, proteins were significantly enriched for nuclear and DNA-related processes and functions, including *nucleosome assembly*, *histone exchange*, *sequence-specific DNA binding* and *nuclear chromosomes*. Impact analysis, which combines classical overrepresentation analysis with the perturbation of a given pathway, demonstrated that *fat and digestion*, *glucagon signalling* and peroxisome proliferator-activated receptor (*PPAR) signalling* pathways were amongst those impacted in iBAT (Figure 3A–D) whilst *RNA degradation, cell adhesion molecules* and *ribosome* pathways were among those impacted in PVAT (Figure 3E–G).

## 4. Discussion

Brown adipose tissue plays a major role in regulating whole body glucose and lipid homeostasis under cold conditions (i.e., when UCP1 is active) with apparent anti-obesity potential in rodents [8]. Here, we demonstrate that when animals are housed at thermoneutrality (i.e., when UCP1 is not active) short-term exposure to a HFD is sufficient to induce rapid whole-body weight gain which may be due to uptake of circulating NEFA. Furthermore, iBAT and PVAT, whilst phenotypically similar, differ in their response to this brief period of nutrient excess. This is the first study to investigate whether these anatomically and developmentally distinct depots [1,2] respond differently to a brief caloric surplus, and suggests that not all BAT is similar with regards to its potential to regulate nutrient metabolism.

An important aspect of our study is the finding that adaptations in iBAT and PVAT with rapid whole-body weight gain occur prior to increased fat mass and it is likely this weight gain is a consequence of lipid accumulation across all fat depots. The changes in iBAT and PVAT could, therefore, be early events in the transition from BAT to a whiter phenotype, the development of adipose tissue dysfunction and /or adaptations in response to caloric surplus. For instance, PSMA3, a component of the core 20S proteasome complex is upregulated in visceral adipose tissue of obese rats [31] and is seen here to be upregulated after only 72h of HFD suggesting a potential role in the acute and chronic adaptation to HFD in these tissues.

In BAT however, activation of the proteasome is essential for cold-induced thermogenesis with selective induction of proteostasis in BAT improving metabolic activity and body weight independent of insulin tolerance in diet-induced obesity [32]. In this study, downregulation of Transmembrane Protein 126a (TMEM126a) in iBAT is of particular interest. It is an inner-mitochondrial, cristae associated transmembrane protein strongly expressed in multiple tissues in adult humans and co-localises with ATP synthase F1 subunit alpha (ATP5A) [33,34]. Intriguingly, TMEM126 also co-localises with, and binds to, a CD137 ligand (CD137L) in macrophages to regulate reverse signaling [35,36]. CD137 is a common marker of beige adipocytes. Whether TMEM126a regulates mitochondrial function in iBAT or in the development of beige adipocytes is unknown. Another mitochondrial protein, Coiled-Coil Domain Containing 51 (CCDC51), has previously been shown to be a target of the transcription factor iroquois homeobox 3 (IRX3). In human obesity, IRX3 is a target of the FTO risk loci with allele carriers having increased IRX3 expression in early adipogenesis where it is proposed to regulate adipocyte function and browning through the modulation of mitochondrial genes [37].

Our finding that the response to brief nutrient excess differs in PVAT compared to iBAT may be explained, in part, by the proximity of PVAT to, and local interaction with, the vascular system. The downregulation of Phosphofructokinase, Platelet (PFKP), which catalyses fructose 6-phosphate to fructose 1,6-bisphophate, is intriguing given that elevated expression is associated with raised BMI and obesity in genome wide association studies [38,39]. In iBAT, however, PFKP expression is induced by cold exposure and sympathetic activation with a β3-agonist and reduced at thermoneutrality [40]. This is in line with the downregulation of both thermogenic and metabolic genes and would suggest perturbed adipocyte function in PVAT. Furthermore, Growth Hormone Inducible Transmembrane Protein (GHITM), a mitochondrial protein involved in cristae organisation, cytochocrome C release and apoptosis was downregulated [41]. Alongside an upregulation of Ubiquitin Conjugating Enzyme E2 N (Ube2n) which regulates the TLR4 signalling pathway, and genes governing the inflammasome it points towards a pro-inflammatory, apoptotic environment in PVAT following only brief exposure to a HFD. Interestingly, Myotrophin (MTPN) and Capping Actin Protein Of Muscle Z-Line Subunit Alpha 1 (CAPZA1), both of which play a role in the growth of actin filaments, were upregulated in PVAT. Of these, MTPN drives the growth of cardiomyocytes and promotes cardiac hypertrophy, whilst reduced CAPZA1 improves post-ischemic cardiac function [42,43]. Whether these proteins in PVAT signal to the endothelium to regulate vascular remodelling is currently unknown.

The enrichment of lipid and cholesterol-related GO terms in iBAT are in accordance with a major role in lipoprotein metabolism [44,45]. Downregulation of proteins involved in reverse-transport of cholesterol from fat to liver and the formation of high-density lipoproteins and chylomicrons (APOA1, 2 and 4) suggests changes in the uptake and processing of triglyceride-rich lipoproteins as fuel for iBAT [46]. Alternatively, and in the context of the rapid whole body weight gain, perturbation of PPAR signaling including reduced UCP1 and upregulation of white adipose adipokines i.e., adiponectin mRNA may indicate early stages of iBAT remodeling towards a white phenotype. In contrast, the enrichment of nuclear related GO terms in PVAT are indicative of dynamic changes in DNA replication, repair and gene transcription [47,48,49]. Whether the genes and proteins in these nuclear-related pathways act on the vascular wall to regulate vascular function following brief exposure to a HFD remains to be determined, as does the extent to which these adaptations can be reversed.

Impact analysis further highlights the divergent response in these two BAT depots with *fat digestion and absorption glucagon signaling* and *PPAR signalling* among those impacted in iBAT. Importantly, downregulation of UCP1 in the PPAR signalling pathway suggests impaired BAT function which may contribute to the rapid weight gain seen within 72h. Furthermore, downregulation of the long-chain fatty-acid CoA ligase 5 (ACSL5) in the PPAR pathway and the Mitochondrial Aspartate Aminotransferase 2 (GOT2), which facilitates cellular long chain fatty acid uptake and metabolite exchange between the cytosol and mitochondria, is significant as long-chain fatty acids activate UCP1 and are the preferred fuel of BAT for adaptive thermogenesis [5]. Any adaptations of this type could modulate NEFA handling and contribute in part to the decline in plasma NEFA with the HFD, although TG were unaffected. Conversely in PVAT, impacted pathways included retinol metabolism, cell adhesion molecules, ribosome and fluid shear stress and atherosclerosis. Retinoic acid regulates adipogenesis and cell migration, differentiation, apoptosis and vascular calcification in vascular smooth muscle cells [50]. A downregulation of Retinol Saturase (RETSAT) may also be indicative of the early stages of PVAT dysfunction. RETSAT knockout mice exhibit increased adiposity due to an upregulation of PPARγ and FABP4 and it is downregulated in obese humans where the infiltration of macrophages represses its function [51,52]. Altered cell adhesion and shear stress pathways in PVAT are intriguing due to their well-known role in driving atherogenesis [53,54]. For instance, the platelet and cell adhesion molecule PECAM1 is essential for vascular remodeling in mice with PECAM1 knockout mice, which are partially protected from atherosclerosis, exhibiting reduced aortic arch and sinus lesions [55,56]. How these proteins in PVAT regulate vascular function is currently unknown but we predict these may be the initial stages of PVAT dysfunction in response to a HFD and, as such, could be important in the initial stages of vascular dysfunction.

## 5. Conclusions

In conclusion, we show that two anatomically and developmentally distinct BAT depots exhibit a divergent response to short-term nutrient excess. We propose these alterations, which occur following rapid weight gain but prior to increased fat mass, are of importance in the development of subsequent adipocyte dysfunction in obesity.

## Figures and Tables

**Figure 1 nutrients-11-01065-f001:**
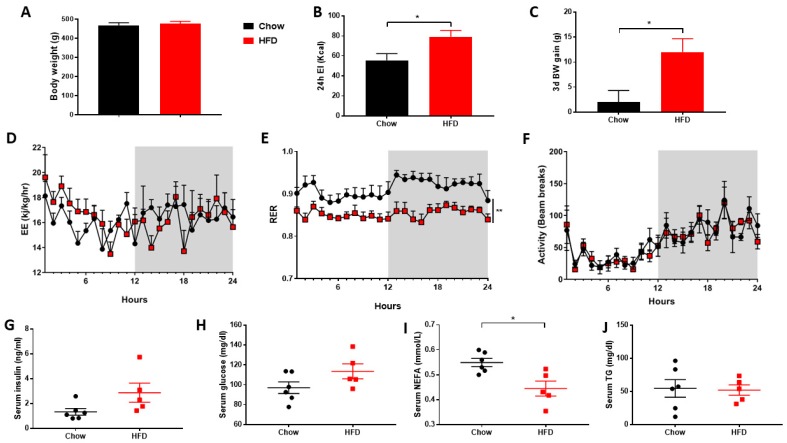
High fat diet (HFD) modified total energy balance but had no effect on insulin, glucose, triglycerides or non-esterified fatty acids. (**A**) Final body weight, (**B**) 24h energy intake, (**C**) 3 day weight gain, (**D**) 24h energy expenditure (EE), respiratory exchange ratio (RER) and ambulatory activity and (**G**–**J**) serum metabolites. Data expressed as mean ± SEM, n = 6 per group. For comparison, data was analysed by either Students *t*-test (**A**–**C**,**G**–**J**) or two-way ANOVA (**D**) and Sidak post-hoc tests. Significance denoted as * < 0.05; ** < 0.01.

**Figure 2 nutrients-11-01065-f002:**
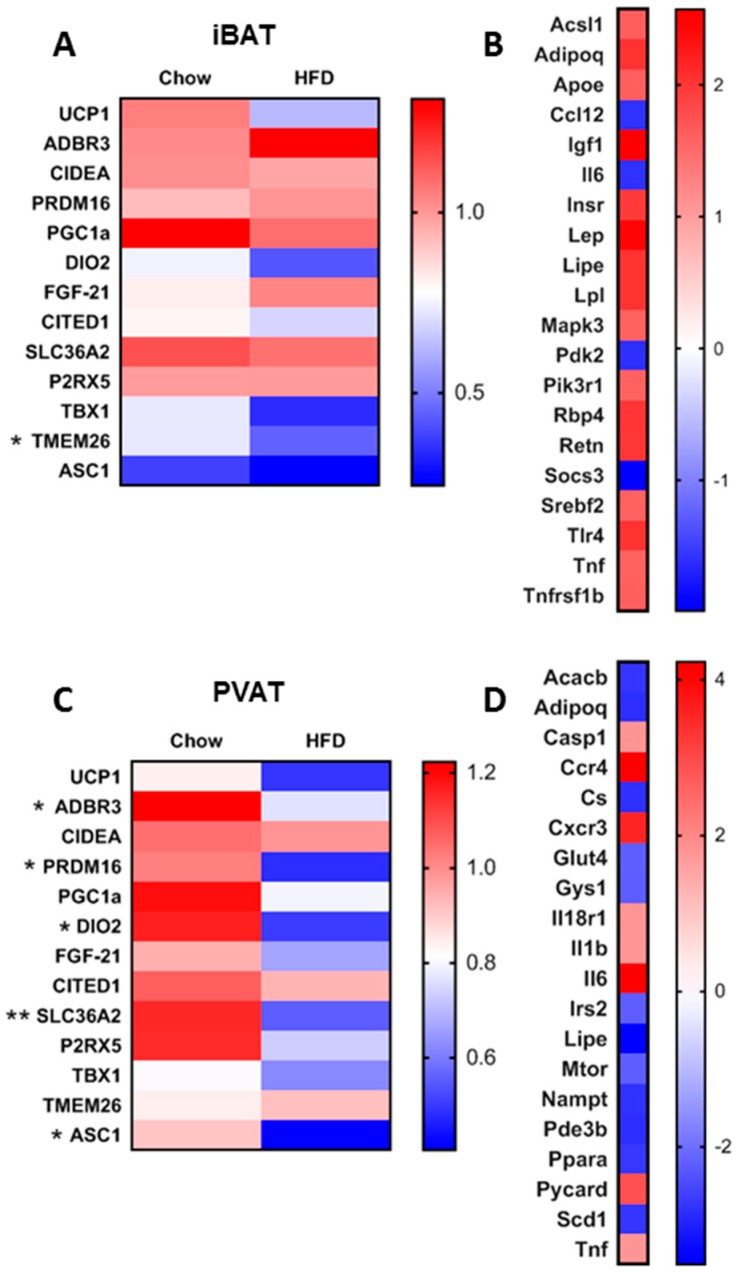
Summary of the effect of the HFD on differences in thermogenic genes involved in brown adipose tissue (BAT), beige and WAT adipocyte function in interscapular BAT (iBAT) (**A**) and perivascular BAT (PVAT) (**C**). Top 20 up/down-regulated genes involved in adipose tissue insulin resistance in iBAT (**B**) and PVAT (**D**). Data expressed as mean (**A**,**C**, n = 5–6) or fold change (**B**,**D**, n = 3). Data were analysed by Students *t*-test (**A**,**C**) with significance denoted as * < 0.05 or ** < 0.01.

**Figure 3 nutrients-11-01065-f003:**
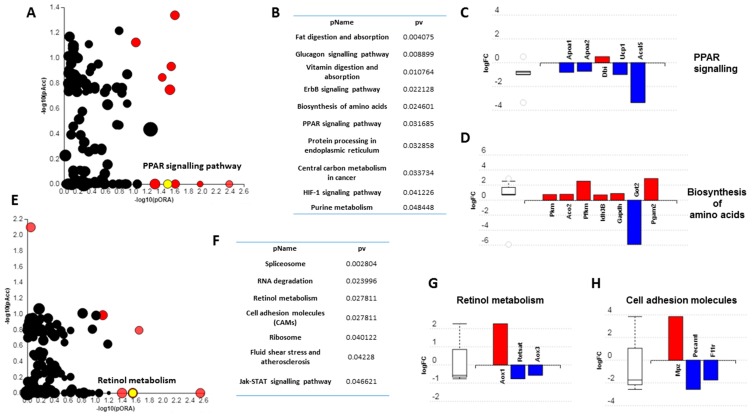
Overview of alterations in the proteome of iBAT and PVAT following 72h of high fat feeding. Impact analysis: iBAT (**A**), PVAT (**E**); Most impacted pathways: iBAT (**B**), PVAT (**F**); Proteins altered in specified pathways: iBAT (**C**,**D**), PVAT (**G**,**H**). Figures created with Advaita Bio IPathwayGuide using only differentially altered proteins. Peroxisome proliferator-activated receptor (PPAR).

**Table 1 nutrients-11-01065-t001:** Top 10 differentially regulated proteins in BAT.

Symbol	Gene Name	Entrez	Logfc	Adjpv
**Psma3l**	Proteasome subunit alpha type-3	408248	**0.793509**	0.000116
**Tmem126a**	Transmembrane protein 126A	293113	**−1.83882**	0.000185
**Ssr3**	Signal Sequence Receptor Subunit 3	81784	**−1.61099**	0.0002
**Ccdc51**	Coiled-Coil Domain Containing 51	316008	**−0.693**	0.000763
**Pkm**	Pyruvate Kinase M1/2	25630	**0.7335**	0.00279
**Vps4a**	Vacuolar Protein Sorting 4 Homolog A	246772	**−0.71726**	0.003363
**Apoa4**	Apolipoprotein A4	25080	**−1.0162**	0.004051
**Prss1**	Serine Protease 1	24691	**0.827668**	0.005684
**Serpina3n**	Serpin Family A Member 3	24795	**−0.58665**	0.006792
**Camk2d**	Calcium/Calmodulin Dependent Protein Kinase II Delta	24246	**−2.55115**	0.007442

Entrez gene ID (Entrez), log fold change (Logfc) where minus symbol equals downregulation, adjusted *P* value (adjPval).

**Table 2 nutrients-11-01065-t002:** Top 10 differentially regulated proteins in PVAT

Symbol	Gene Name	Entrez	Logfc	Adjpv
**Pfkp**	Phosphofructokinase, Platelet	60416	**−1.38584**	0.000225
**Hdgf**	Heparin Binding Growth Factor	114499	**0.69362**	0.000445
**Rbmxrtl**	RNA-binding motif protein, X chromosome retrogene-like	307779	**1.91652**	0.001389
**Ywhaq**	14-3-3 Protein Theta	25577	**0.613265**	0.001405
**Ghitm**	Growth Hormone Inducible Transmembrane Protein	290596	**−0.71467**	0.001854
**Capza1**	Capping Actin Protein Of Muscle Z-Line Subunit Alpha 1	691149	**1.194102**	0.002081
**Mtpn**	Myotrophin	79215	**0.669685**	0.002234
**Ube2n**	Ubiquitin Conjugating Enzyme E2 N	116725	**0.80585**	0.002495
**B2m**	Beta-2- Microglobulin	24223	**0.809804**	0.002742
**Acox1**	Acyl-CoA Oxidase 1	50681	**−3.50641**	0.004222

Entrez gene ID (Entrez), log fold change (Logfc) where minus symbol equals downregulation, adjusted *P* value (adjPval).

**Table 3 nutrients-11-01065-t003:** GO terms enriched in BAT.

**GoId**	**GoName**	**CountDE**	**CountAll**	**Pv_elim**
**Biological Process**				
**GO:0039536**	negative regulation of RIG-I signaling pathway	3	3	0.0011
**GO:0050996**	positive regulation of lipid catabolic process	4	6	0.0014
**GO:0046470**	phosphatidylcholine metabolic process	5	7	0.0038
**GO:0030300**	regulation of intestinal cholesterol absorption	3	4	0.004
**GO:0034380**	high-density lipoprotein particle assembly	3	4	0.004
**Molecular Function**				
**GO:0031210**	phosphatidylcholine binding	3	3	0.0011
**GO:0060228**	phosphatidylcholine-sterol O-acyltransferase activator activity	3	4	0.004
**GO:0003713**	transcription coactivator activity	4	8	0.0054
**GO:0001047**	core promoter binding	3	5	0.0091
**GO:0017127**	cholesterol transporter activity	3	5	0.0091
**Cellular Component**				
**GO:0034366**	spherical high-density lipoprotein particle	3	4	0.0041
**GO:0042627**	chylomicron	3	4	0.0041
**GO:0005667**	transcription factor complex	3	6	0.0174
**GO:0034361**	very-low-density lipoprotein particle	3	6	0.0174

**Table 4 nutrients-11-01065-t004:** GO terms enriched in PVAT.

**GoId**	**GoName**	**CountDE**	**CountAll**	**Pv_elim**
**Biological Process**				
**GO:0006334**	nucleosome assembly	8	10	0.00023
**GO:0017144**	drug metabolic process	5	8	0.00287
**GO:0043486**	histone exchange	3	3	0.00343
**GO:1901655**	cellular response to ketone	8	21	0.00825
**GO:0021766**	hippocampus development	6	14	0.0116
**Molecular Function**				
**GO:0042393**	histone binding	7	8	0.000011
**GO:0003785**	actin monomer binding	3	3	0.0033
**GO:0043565**	sequence-specific DNA binding	8	23	0.0143
**GO:0035091**	phosphatidylinositol binding	5	8	0.0219
**GO:0005506**	iron ion binding	6	16	0.0227
**Cellular Component**				
**GO:0000788**	nuclear nucleosome	3	3	0.0035
**GO:0000784**	nuclear chromosome, telomeric region	3	4	0.0123
**GO:0030054**	cell junction	38	183	0.0174
**GO:0071013**	catalytic step 2 spliceosome	5	12	0.0246
**GO:0001931**	uropod	3	5	0.0273

## Data Availability

The datasets used and analysed during the current study are available from the corresponding author on reasonable request.

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
