# Peer review of "Interscapular and Perivascular Brown Adipose Tissue Respond Differently to a Short-Term High-Fat Diet"

_nutrients, 2019, doi:10.3390/nu11051065_

Round 1

Reviewer 1 Report

Aldiss P. and coworkers present the manuscript entitled ‘Interscapular and perivascular brown adipose tissue respond differently to a short-term high-fat diet’ in which it is demonstrated that only 72 hours of high fat diet exposure induces important changes in the thermogenic profile of perivascular brown fat to white-like-phenotype, while not in interscapular one. Authors screen several genes, and stablish a good interpretation of the changes observed. However, I have some concerns that need to be addressed.

1)    It is really surprising the profound changes that occur in such a short time of high fat diet, but it would be very interesting to know if these changes could be easily reversed through short time of standard diet (72 h too) or conversely, reversion of deleterious effects on PVAT require longer healthy feeding. The quick reversion or not in PVAT should be explored.

2)    iBAT is mainly involved to thermogenesis and energy homeostasis, PVAT activity is a key regulator of vascular function. However author only explore the metabolic parameters, but nothing about the vascular reactivity, since maybe PVAT changes could induce endothelial dysfunction after 72 h HFD. This possibility must be tested.

Author Response

Aldiss P. and coworkers present the manuscript entitled ‘Interscapular and perivascular brown adipose tissue respond differently to a short-term high-fat diet’ in which it is demonstrated that only 72 hours of high fat diet exposure induces important changes in the thermogenic profile of perivascular brown fat to white-like-phenotype, while not in interscapular one. Authors screen several genes, and stablish a good interpretation of the changes observed. However, I have some concerns that need to be addressed.

1)    It is really surprising the profound changes that occur in such a short time of high fat diet, but it would be very interesting to know if these changes could be easily reversed through short time of standard diet (72 h too) or conversely, reversion of deleterious effects on PVAT require longer healthy feeding. The quick reversion or not in PVAT should be explored.

Our response: This is an interesting suggestion which we have added to the discussion (see lines 270-3) but is beyond the scope of the current manuscript. Moreover, it should be noted the project from the British Heart Foundation for the research presented has now been completed, and thus neither the personal or research funds to undertake further extensive studies are available. It would thus seem reasonable to publish the present work and allow others to capitalise on our main findings, as appropriate.

2)    iBAT is mainly involved to thermogenesis and energy homeostasis, PVAT activity is a key regulator of vascular function. However author only explore the metabolic parameters, but nothing about the vascular reactivity, since maybe PVAT changes could induce endothelial dysfunction after 72 h HFD. This possibility must be tested.

Our response: As noted in our response this is an interesting suggestion which we have added to the discussion (see lines 270-3) but is beyond the scope of the current manuscript. Moreover, it should be noted the project from the British Heart Foundation for the research presented has now been completed, and thus neither the personal or research funds to undertake further extensive studies are available. It would thus seem reasonable to publish the present work and allow others to capitalise on our main findings, as appropriate.

Reviewer 2 Report

Aldis et al, aimed to determine whether iBAT and PVAT differ in their response to a short-term caloric surplus. We think that the results could be interesting, however, the way the authors present their data is too descriptive and scientific rationale is lacking.

What is the rationale of 72h of DIO ? In the introduction part, the authors state that “after only 24h of a high-fat diet (HFD) with central and peripheral insulin resistance, adipose tissue inflammation and hepatic steatosis occurring within 3-4 days”. Confusing. Please explain and justify the scientific rationale.

PVAT isolation procedure is needed. Which location was chosen ? Why ? PVAT weights are missing and are of course needed (ND vs HDF)

Such sentence is not clear “As expected, there was no difference between groups with regard to final body weight, total fat mass…” since 72h after DIO BW gain is indeed higher in DIO group. Does it mean that ND vs HFD groups had different basal body weights? Confusing. Clarify. Body weight evolution, day by day is also needed. In the same way, in the discussion section please clarify this “rapid whole-body weight gain occur prior to increased fat mass and it is likely this weight gain is a consequence of lipid accumulation across all fat depots”. Since various fat depots were not different between both groups, how do you explain weight gain. Once again this is confusing for the reader.

When did EI measurement was performed. This information is requested and seems absent in the method section. EE display similar profiles. Please provide AUC total and per light period with statistics.

Serum insulin must be presented as dots. It seems that there are/is some outlier(s). Only 6 rats were used in each group. This is to our point of view one of the very weak point of the study, hence outliers must be shown and data reanalyzed depending on the outliers. NEFA levels decrease in HFD group and the authors do not discuss and justify such results. This must be added and explained. TG are not modified as well, please explain, comment and clarify.

To our point of view, data must be reanalysed or presented in a different way. For instance, UCP1 levels seem different in IBAT vs PVAT in chow diet. Is it possible to compare Figure 2A and C? It would be interesting to compare Δ (chow vs HDF) between IBAT and PVAT. Similarly, in figure 2C, UCP1 looks downregulated by HFD, though the authors claim that it is not (“Despite similar UCP1 mRNA”). Once again they must completely reconsider their data presentation. Moreover it seems that PVAT is “more” brown than IBAT in Chow diet animals? What is the impact of thermoneutrality on PVAT ? This must be analysed and commented in the manuscript.

Author Response

Aldis et al, aimed to determine whether iBAT and PVAT differ in their response to a short-term caloric surplus. We think that the results could be interesting, however, the way the authors present their data is too descriptive and scientific rationale is lacking.

What is the rationale of 72h of DIO ? In the introduction part, the authors state that “after only 24h of a high-fat diet (HFD) with central and peripheral insulin resistance, adipose tissue inflammation and hepatic steatosis occurring within 3-4 days”. Confusing. Please explain and justify the scientific rationale.

Our response: This sentence has been revised accordingly.

PVAT isolation procedure is needed. Which location was chosen ? Why ? PVAT weights are missing and are of course needed (ND vs HDF)

Our response: We have added further details of the PVAT dissection as requested on line 81. Moreover, it is not standard practise to weigh the depot. The dissection is a delicate process where avoiding vascular and connective tissue is paramount and as such, sampled amounts will undoubtedly vary between animals.

Such sentence is not clear “As expected, there was no difference between groups with regard to final body weight, total fat mass…” since 72h after DIO BW gain is indeed higher in DIO group. Does it mean that ND vs HFD groups had different basal body weights? Confusing. Clarify. Body weight evolution, day by day is also needed. In the same way, in the discussion section please clarify this “rapid whole-body weight gain occur prior to increased fat mass and it is likely this weight gain is a consequence of lipid accumulation across all fat depots”. Since various fat depots were not different between both groups, how do you explain weight gain. Once again this is confusing for the reader.

Our response: We have revised this section of the results to improve clarity – see lines 152=155. The anaimls were only weighed at the start and end of the study.

When did EI measurement was performed. This information is requested and seems absent in the method section. EE display similar profiles. Please provide AUC total and per light period with statistics.

Our response: As described in the Methods, EI was performed for the last 24 h of the study – see line 76. We have added EE for light and dark phases respectively which shows there is no difference between groups – see Suppl Figure 1.

Serum insulin must be presented as dots. It seems that there are/is some outlier(s). Only 6 rats were used in each group. This is to our point of view one of the very weak point of the study, hence outliers must be shown and data reanalyzed depending on the outliers. NEFA levels decrease in HFD group and the authors do not discuss and justify such results. This must be added and explained. TG are not modified as well, please explain, comment and clarify.

Our response: 1G,H,I and J amended accordingly and shows no effect of any outliers. Discussion amended as requested – see lines 284-5.

To our point of view, data must be reanalysed or presented in a different way. For instance, UCP1 levels seem different in IBAT vs PVAT in chow diet. Is it possible to compare Figure 2A and C? It would be interesting to compare Δ (chow vs HDF) between IBAT and PVAT. Similarly, in figure 2C, UCP1 looks downregulated by HFD, though the authors claim that it is not (“Despite similar UCP1 mRNA”). Once again they must completely reconsider their data presentation. Moreover it seems that PVAT is “more” brown than IBAT in Chow diet animals? What is the impact of thermoneutrality on PVAT ? This must be analysed and commented in the manuscript.

Our response: The gene array data can only be compared for each depot between dietary groups as none of the reference genes were stable between the two depots. The figure title has been amended accordingly. The Figures thus illustrate the differences due to diet alone, and cannot be used to compare across depots. The effect of housing temperature per se was not part of this present study and thus cannot be directly commented upon. 

Reviewer 3 Report

The manuscript by Aldiss et al is an interesting study on the effects of a short-term high-fat diet on interscapular  brown adipose tissue (iBAT) and perivascular adipose tissue (PVAT). PVAT has been reported to be phenotypically similar to BAT, although having different developmental origin and physiological role. For the first time, the present work analyses by gene expression and proteomics the effects of an acute nutrient stress on both ATs in rats housed at thermoneutrality. Results indicated striking differences in iBAT and PVAT thermogenic and insulin responsiveness-related gene expression in response to a short-term HFD. In addition, the well-performed proteomic study also contributes to stress that both ATs respond differently to this acute nutritional stress. Although this is a descriptive study, overall data are novel and of interest. This reviewer has only a few comments and some specific points that need to be addressed.

 -Methods, line 101. Probe validation data of Insulin Resistance (SAB target list) PCR Array (BioRad) is stated to be available in supplementary data. However, this reviewer has not found them. Regarding the corresponding results reported in Fig 2 B and D, there is a different list of genes reported for each AT depot. Why? Are only genes with significantly altered expression shown? This has to be explained.

 -Results, line 152. Delete the reference to insulin signalling in this heading 3.1.

 -Results, lines 181-187. Please indicate whether there is an up- or down-regulation of the referred proteins. Moreover, it would be useful to have legends in the tables explaining the column headings. In Tables 1 and 2, indicate also the meaning of the minus sign (down-regulation).  

 -Fig 3. Are they statistically significant the protein changes depicted in panels C, D, G and H?

 -In general, the manuscript is clearly written but perhaps excessively concise. For example, in the Discussion section, it is difficult to follow some reasoning because the results analyzed have not been previously explained in the Results section. It will help to indicate the Fig or the Table were the referred results are reported. Moreover, it would be useful to better describe some of the proteins encoded by the referred genes (i.e., GOT2 is the mitochondrial aspartate aminotransferase; ACSL5 is the long chain fatty acid CoA ligase 5, and it is an acyl CoA synthetase isoenzyme; RETSAT is retinol saturase, etc).

 -Discussion, line 235. That PSMA3 is found to be upregulated in iBAT (Table 1) has to be indicated.

 -Discussion, line 256. Add that the sentence refers to adipocyte function in PVAT.

 -Typos: line 222, thermoeneutrality; Table 2, fkp is Pfkp

Author Response

The manuscript by Aldiss et al is an interesting study on the effects of a short-term high-fat diet on interscapular  brown adipose tissue (iBAT) and perivascular adipose tissue (PVAT). PVAT has been reported to be phenotypically similar to BAT, although having different developmental origin and physiological role. For the first time, the present work analyses by gene expression and proteomics the effects of an acute nutrient stress on both ATs in rats housed at thermoneutrality. Results indicated striking differences in iBAT and PVAT thermogenic and insulin responsiveness-related gene expression in response to a short-term HFD. In addition, the well-performed proteomic study also contributes to stress that both ATs respond differently to this acute nutritional stress. Although this is a descriptive study, overall data are novel and of interest. This reviewer has only a few comments and some specific points that need to be addressed.

 -Methods, line 101. Probe validation data of Insulin Resistance (SAB target list) PCR Array (BioRad) is stated to be available in supplementary data. However, this reviewer has not found them. Regarding the corresponding results reported in Fig 2 B and D, there is a different list of genes reported for each AT depot. Why? Are only genes with significantly altered expression shown? This has to be explained.

Our response: Regarding the results in Fig. 2 B and D, as stated in the legend - the heatmap represents the top 20 differentially regulated in each specific tissue. Representative validation data is now included available in the supplementary data (Suppl Fig. 2.). We validated two genes from this study both of which were significant and showed the same effect as on the array plate. Additional genes from other plates/studies corroborate the array plates so we can be certain they are accurate.

 -Results, line 152. Delete the reference to insulin signalling in this heading 3.1.

Our response: Deleted as suggested

 -Results, lines 181-187. Please indicate whether there is an up- or down-regulation of the referred proteins. Moreover, it would be useful to have legends in the tables explaining the column headings. In Tables 1 and 2, indicate also the meaning of the minus sign (down-regulation).  

Our response: Arrows to indicate up or downregulation have been added after each protein and legends have been added to explain headings.

 -Fig 3. Are they statistically significant the protein changes depicted in panels C, D, G and H?

Our response: Yes, all GO and pathway analysis was performed using only the list of differentially regulated proteins. This has been clarified in the legend of Fig. 3.

 -In general, the manuscript is clearly written but perhaps excessively concise. For example, in the Discussion section, it is difficult to follow some reasoning because the results analyzed have not been previously explained in the Results section. It will help to indicate the Fig or the Table were the referred results are reported. Moreover, it would be useful to better describe some of the proteins encoded by the referred genes (i.e., GOT2 is the mitochondrial aspartate aminotransferase; ACSL5 is the long chain fatty acid CoA ligase 5, and it is an acyl CoA synthetase isoenzyme; RETSAT is retinol saturase, etc).

Our response: To improve clarity we have either describer or added full protein names.

 -Discussion, line 235. That PSMA3 is found to be upregulated in iBAT (Table 1) has to be indicated.

Our response: We have indicated that it is upregulated

 -Discussion, line 256. Add that the sentence refers to adipocyte function in PVAT.

Our response: We have added this.

 -Typos: line 222, thermoeneutrality; Table 2, fkp is Pfkp

Our response: We have corrected these.

Round 2

Reviewer 1 Report

I think that the manuscript is ready to be published in Nutrients.

Author Response

No response required.

Reviewer 2 Report

The authors did not take into consideration any of the reviewer's comments and to me the manuscript has not been improved by their minor changes and the very short answers they proposed.

Author Response

No response required.